# Impact of Physical Activity on the Characteristics and Metabolic Consequences of Alcohol Consumption: A Cross-Sectional Population-Based Study

**DOI:** 10.3390/ijerph192215048

**Published:** 2022-11-15

**Authors:** Onni Niemelä, Aini Bloigu, Risto Bloigu, Anni S. Halkola, Markus Niemelä, Mauri Aalto, Tiina Laatikainen

**Affiliations:** 1Department of Laboratory Medicine, Medical Research Unit, Seinäjoki Central Hospital and Tampere University, 60220 Seinäjoki, Finland; 2Center for Life Course Health Research, University of Oulu, 90570 Oulu, Finland; 3Infrastructure for Population Studies, Faculty of Medicine, University of Oulu, 90570 Oulu, Finland; 4Department of Anesthesiology, Oulu University Hospital, 90220 Oulu, Finland; 5Department of Psychiatry, Seinäjoki Central Hospital and Tampere University, 33100 Tampere, Finland; 6Department of Public Health and Social Welfare, Finnish Institute for Health and Welfare (THL), 00271 Helsinki, Finland; 7Institute of Public Health and Clinical Nutrition, University of Eastern Finland, 70210 Kuopio, Finland; 8Joint Municipal Authority for North Karelia Social and Health Services, 80210 Joensuu, Finland

**Keywords:** ethanol, lifestyle, obesity, physical exercise, smoking

## Abstract

Sedentary lifestyle and excessive alcohol drinking are major modifiable risk factors of health. In order to shed further light on the relationships between physical activity and health consequences of alcohol intake, we measured biomarkers of liver function, inflammation, lipid status and fatty liver index tests in a large population-based sample of individuals with different levels of physical activity, alcohol drinking and other lifestyle risk factors. The study included 21,050 adult participants (9940 men, 11,110 women) (mean age 48.2 ± 13.3 years) of the National FINRISK Study. Data on physical activity, alcohol drinking, smoking and body weight were recorded. The participants were classified to subgroups according to gender, levels of physical activity (sedentary, low, moderate, vigorous, extreme), alcohol drinking levels (abstainers, moderate drinkers, heavy drinkers) and patterns (regular or binge, types of beverages preferred in consumption). Serum liver enzymes (GGT, ALT), C-reactive protein (CRP) and lipid profiles were measured using standard laboratory techniques. Physical activity was linearly and inversely related with the amount of alcohol consumption, with the lowest alcohol drinking levels being observed in those with vigorous or extreme activity (*p* < 0.0005). Physically active individuals were less frequently binge-type drinkers, cigarette smokers or heavy coffee drinkers than those with sedentary activity (*p* < 0.0005 for linear trend in all comparisons). In the General Linear Model to assess the main and interaction effects of physical activity and alcohol consumption on biomarker status, as adjusted for anthropometric measures, smoking and coffee consumption, increasing levels of physical activity were found to be associated with more favorable findings on serum GGT (*p* < 0.0005), ALT (*p* < 0.0005 for men), cholesterol (*p* = 0.025 for men; *p* < 0.0005 for women), HDL-cholesterol (*p* < 0.0005 for men, *p* = 0.001 for women), LDL-cholesterol (*p* < 0.03 for men), triglycerides (*p* < 0.0005 for men, *p* < 0.03 for women), CRP (*p* < 0.0005 for men, *p* = 0.006 for women) and fatty liver index (*p* < 0.0005). The data support the view that regular moderate to vigorous physical activity may counteract adverse metabolic consequences of alcohol consumption on liver function, inflammation and lipid status. The role of physical activity should be further emphasized in interventions aimed at reducing health problems related to unfavorable risk factors of lifestyle.

## 1. Introduction

Lack of physical exercise is a lifestyle risk factor, which has been globally recognized as an increasingly common contributor to the incidence of chronic diseases and premature death [1,2,3,4]. Previous studies have also suggested synergistic interactions between physical inactivity and other unfavorable factors of lifestyle, such as alcohol drinking, smoking or excess body weight [3,4,5]. Therefore, interventions aimed at reducing the burden of modifiable lifestyle risk factors have also been emphasized as a major focus for personalized medicine and public health policies [6,7]. The relationships between various types of exposures and health responses in individuals with combinations of lifestyle factors have, however, remained poorly known. While regular physical exercise is expected to provide a wealth of health benefits [8,9,10,11], the intensities of activity leading to beneficial or sometimes adverse consequences have also remained as a subject of debate [12,13,14,15,16,17].

A large body of recent evidence has accumulated indicating that the adverse metabolic consequences of unhealthy lifestyle are reflected in distinct biomarkers of liver function, lipid status and inflammation [18,19,20,21,22]. The early changes in such markers even within their normal ranges may predict both hepatic and extra-hepatic disease risks, including metabolic syndrome, and cardio- or cerebrovascular events [21,23,24]. While the biochemical pathways underlying such observations have remained unclear, the biomarker-based studies so far have indicated significant roles of inflammation and oxidative stress in the sequence of events leading from lifestyle risk factor exposure to tissue damage.

As yet, only limited information has been available on the effect of physical activity as a mediator of changes on biochemical indices of health in individuals presenting with distinct lifestyle risk factor combinations. We investigated here how different levels of physical activity are represented in apparently healthy individuals with various levels and patterns of alcohol drinking, smoking, coffee consumption or overweight in a large national population-based material of the FINRISK Study. The impacts of physical activity on health were also assessed using measurements of biomarkers of liver status (ALT, GGT), inflammation (C-reactive protein), lipid metabolism (cholesterol, HDL-cholesterol, LDL-cholesterol, triglycerides) and fatty liver index (FLI, a predictor of fatty change in the liver). It is assumed that further understanding of the interactions with the various determinants of lifestyle may improve our possibilities for interventions aimed at adopting more favorable lifestyles.

## 2. Materials and Methods

### 2.1. Study Design, Data Sources and Participants

Data were collected from a large cross-sectional population health survey representing an age- and gender stratified random sample from the population register in Finland (The National FINRISK Study) in years 1997, 2002 and 2007 [5,25]. The clinical assessments included anthropometric measures, laboratory data and questionnaires collecting structured information on current health status and medical history, physical activity, alcohol intake, smoking as well as coffee consumption [25,26]. The questionnaires used designated the responses to each question on various lifestyle determinants to distinct categories, as previously described [5,26,27]. The data for the present study were available from 21,050 apparently healthy individuals: 9940 men (47.2%) and 11,110 women (52.8%) (mean age 48.2 ± 13.3 years, range 25–74 years) who completed the questionnaires and participated in the medical examinations. Individuals with any apparent history or current clinical signs of liver disease, ischemic heart or brain disease or active infection at the time of blood sampling were excluded.

The data for habitual physical activity included both the number and total time used for leisure-time physical exercises with intensity leading to shortness of breath or sweating. In addition, physical activity in work time was recorded. The sum of these provided the total time engaged in physical activity. The study population was subsequently classified to subgroups as follows:Sedentary activity, less than 15 min per day or no activity (681 men, 303 women);Low activity, 15–45 min of physical activity per day (917 men, 676 women);Moderate activity, 45–90 min of physical activity per day (5495 men, 5971 women);Vigorous activity, 90–120 min of physical activity per day (1814 men, 3033 women);Extreme activity, over 120 min of physical activity per day (1033 men, 1127 women).

Data on alcohol consumption were based on self-reports covering the period of 12 months prior to blood sampling. The data gathered included information on the total amounts of ethanol-containing drinks, frequencies of consumption and the types of beverages consumed [5]. The ethanol content in different beverages was estimated in grams of ethanol based on defined portion sizes as follows: wine 12 g (12 cL), regular beer 12 g (1/3 L), strong beer 15.5 g (1/3 L), cider 12 g (1/3 L), long drink 15.5 g (1/3 L) and spirit 12 g (4 cL). Those consuming alcohol were further classified according to the pattern of drinking to either regular drinkers or binge-type drinkers [28]. Based on the data on total regular alcohol consumption the subjects were categorized to subgroups as follows: i: abstainers, ii: moderate drinkers (≤7 drinks per week for women or ≤14 drinks per week for men) or iii: heavy drinkers, who exceeded 7 drinks per week (women) or 14 drinks per week (men) [29]. Binge drinking was defined as a pattern, which typically had consisted of occasional heavy bouts of drinking at least once a month in amounts exceeding 60 g of alcohol for men or 40 g of alcohol for women on each occasion, which typically leads to blood alcohol levels above 0.8 per mill [30,31]. The participants were also classified according to the type of alcoholic beverage preferred (>50% of total consumption) into the following categories (wine, beer, cider or long drink, spirit or mixed type). The mixed group consisted of individuals in whom none of the above specific types of beverages exceeded 50% of the total consumption.

Body weight and height were measured to the nearest 0.1 kg and 0.1 cm, respectively. Body mass index (BMI, kg/m^2^) was determined as a measure of relative body weight. Waist circumference was measured to the nearest 0.5 cm between the lowest rib and the iliac crest while exhaling.

Information on smoking habits and coffee consumption were recorded with standardized questionnaires as described previously [5]. The data collected were expressed as the number of cigarettes per day or as the intake of standard cups of coffee per day, respectively.

The study was approved by the Coordinating Ethics Committee of the Helsinki and Uusimaa Hospital District in 2002 and 2007 and from the Ethics Committee of the National Public Health Institute in 1997 (1997:38/96; 2002:87/2001; 2007:229/EO/2006). All surveys were conducted in accordance with the Declaration of Helsinki and the ethical rules of the National Public Health Institute.

### 2.2. Laboratory Analyses

Serum alanine aminotransferase (ALT) and gamma-glutamyltransferase (GGT) were analyzed using standard clinical chemical methods on Abbott Architect clinical chemistry analyzer (Abbott Laboratories, Abbott Park, IL, USA). The concentrations of high-sensitivity CRP, a biomarker of inflammation, was determined using a latex immunoassay (Sentinel Diagnostics, Milan, Italy) with the Abbott Architect c8000 immunochemistry analyzer. Lipid profiles including determinations of total cholesterol, high-density lipoprotein-associated cholesterol (HDL), low-density lipoprotein (LDL) and total triglycerides were based on standard enzymatic methods. The cut-offs for the normal limits of the current laboratory markers were as follows: ALT (50 U/L men; 35 U/L women), GGT (60 U/L men; 40 U/L women), CRP (3.0 mg/L), cholesterol (5 mmol/L), HDL cholesterol (1.0 mmol/L men, 1.2 mmol/L women), LDL cholesterol (3.0 mmol/L), triglycerides (1.7 mmol/L). Fatty liver index, a predictor algorithm for fatty liver disease, was computed based on BMI, waist circumference, triglycerides and GGT, as previously described [32,33].

### 2.3. Statistical Methods

The data are presented as means and standard deviations (SDs) or frequencies and percentages, as indicated. The ANOVA trend test and the Chi-square test for trend were conducted to evaluate the trends in continuous and categorical variables, respectively. The General Linear Model (GLM) with ordinary least squares (OLS) was applied to assess the main and interaction effects of physical activity and alcohol consumption on various biomarkers using BMI, smoking status and coffee consumption as covariates. Prior to analysis, a logarithmic transformation was performed for biomarkers with skewed distribution. After analysis, the estimates were back-transformed to represent their original scale. A *p*-value < 0.05 was considered statistically significant. The analyses were carried out with IBM SPSS Statistics 28.0 (Armonk, NY, USA: IBM Corp.).

## 3. Results

The main demographic characteristics of the participants in subgroups classified according to the different levels of physical activity and gender are summarized in Table 1. Age of the study subjects showed a linear or quadratic association between physical activity such that for men the highest mean ages were noted in those with the highest levels of activity (*p* < 0.0005). In women, the youngest participants represented the middle portion of the intensities (*p* < 0.0005). For both men and women, a linear decreasing trend was noted between the indices of body weight and physical activity (*p* < 0.0005).

The data on alcohol consumption and other determinants of lifestyle in the subgroups of individuals with different levels of physical activity are summarized in Table 2. The subjects with alcohol consumption were classified to subgroups based on differences in average drinking levels (abstainers, moderate drinkers, heavy drinkers), patterns of drinking (regular or binge) and differences in the dominant type of alcohol preferred. Average levels of drinking were found to be inversely related with physical activity such that the lowest proportions of heavy drinkers and moderate drinkers in men were noted in those with the highest levels of physical activity (*p* < 0.0005 for all comparisons). Binge-type drinking was also significantly less common in physically active individuals (*p* < 0.0005 for both men and women). A linear inverse association also existed between physical activity and beer drinking (*p* < 0.05), smoking status (*p* < 0.0005) and coffee consumption in men (*p* < 0.0005). For women, a quadratic association for coffee consumption was noted so that the lowest levels were observed in those with moderate levels of activity (*p* < 0.0005). In women, consumption of wine was found to be more typical in physically active individuals than in those with sedentary activity (*p* < 0.05).

Figure 1 summarizes the data on laboratory markers of liver function, lipid status, inflammation and fatty liver index (FLI) in individuals with different ordered levels of physical activity and alcohol consumption in men (Figure 1a) and in women (Figure 1b). Unfavorable biomarker status was found to be most common in those with alcohol consumption together with sedentary or low levels of physical activity. In the General Linear Model to assess the main and the interaction effects of physical activity and alcohol consumption, as adjusted for BMI, smoking status and coffee consumption, significant effects of alcohol consumption were noted for GGT (*p* < 0.0005), ALT (*p* < 0.0005), total cholesterol (*p* < 0.0005 for men, *p* = 0.003 for women), HDL-cholesterol (*p* < 0.0005), LDL-cholesterol (*p* = 0.005 for women), triglycerides (*p* < 0.04 for men, *p* < 0.0005 for women), CRP (*p* < 0.04 for men, *p* < 0.03 for women) and FLI (*p* < 0.0005). Independent of alcohol consumption, increasing levels of physical activity were found to be associated with significant favorable changes in serum GGT (*p* < 0.0005), ALT (*p* < 0.0005 for men), total cholesterol (*p* < 0.03 for men, *p* < 0.0005 for women), HDL-cholesterol (*p* < 0.0005 for men, *p* = 0.001 for women), LDL-cholesterol (*p* < 0.03 for men), triglycerides (*p* < 0.0005 for men, *p* < 0.03 for women), CRP (*p* < 0.0005 for men, *p* = 0.006 for women) and FLI (*p* < 0.0005). In women, physical activity was found to lead to a more marked improvement in GGT and FLI among heavy drinkers than in other alcohol consuming groups (*p* < 0.001 and *p* = 0.002 for two-factor interactions, respectively). In additional comparisons of the patterns of responses between men and women, serum LDL and total cholesterol responses were found to show differences in those with the highest levels of physical activity, whereas in other markers the magnitude and direction of changes were relatively similar (Figure 1a,b).

## 4. Discussion

The present study among a large national population sample of apparently healthy individuals indicate that physical activity is a significant determinant of lifestyle, which may also modulate the health outcomes associated with alcohol drinking. Through different levels and patterns of alcohol consumption, the physically active individuals were found to be less susceptible for abnormalities in biomarkers of liver status, inflammation, lipid profiles and FLI indicating that significant health benefits could be achieved by taking regular physical exercise in individuals presenting with unfavorable factors of lifestyle. Our data also emphasize the usefulness of laboratory tests as outcome measures during interventions aimed at low-risk lifestyle.

Having a sedentary lifestyle is globally recognized as a cause for many chronic diseases [3,5,8,9,10,34,35,36,37]. The present data show that sedentary lifestyle is also significantly associated with higher levels of alcohol drinking, another high-risk factor of lifestyle. On the other hand, individuals with moderate to vigorous levels of physical exercise were found to show significantly lower levels of alcohol drinking, supporting the view that exercise could probably also be used as a tool for combatting the habit of heavy drinking, which frequently leads to alcohol addiction and associated medical disorders. In line with this view, recent studies have indicated that the negative effects of stress, a major environmental element of susceptibility to mood disorders, can be alleviated by physical activity [11,38]. Of note, physically active individuals here also presented with a markedly lower prevalence of binge-type drinking, which is currently considered a predominant risk factor for the development of alcohol use disorders in alcohol consumers [39,40,41].

The present findings further indicate that moderate to vigorous physical activity in alcohol consumers may decrease the risk of fatty liver, as assessed using a previously designed predictor algorithm, FLI [32]. In earlier studies, vigorous physical activity was found to reduce hepatic fat content, inflammation and oxidative stress even prior to any measurable changes in body weight of patients with hepatic steatosis [42,43]. Thus, it appears that lifestyle-associated liver burden could be significantly reduced by taking regular physical exercises. Recent studies have also indicated that physical activity is associated with longer life expectancies across all levels of overweight when compared to individuals with sedentary activity supporting also long-term health benefits [10,34,44,45,46,47,48].

The present findings also underscore the importance of avoiding combinations of unfavorable factors. Previous studies have shown that in individuals presenting with adiposity even modest levels of alcohol drinking increase the risk of hepatotoxicity [5,19,20,38,49,50]. Such synergistic effects also exist between smoking and alcohol use [51,52]. Joint pathophysiological pathways involved in diseases driven by the unfavorable lifestyle factors and their combinations seem to be aberrations in the status of inflammation, oxidative stress and fatty acid metabolism [42,53]. Recent interventions targeted towards lifestyle changes among high-risk individuals have been shown to be beneficial in a wide variety of conditions, including liver and heart diseases, diabetes or cancer [3,7,35,42,54,55,56]. Taking physical exercise in a regular manner as part of treatment protocols may be expected to lead to more favorable metabolic profiles with associated reduction in the levels of predictive biomarkers of health [18,24,57,58]. Physical activity seems to play a role in correct maintenance of GGT enzyme, which based on previous observations can be considered a marker of oxidative stress and a mechanistic link between hepatic and extrahepatic disease manifestations due to its pathogenic role in LDL oxidation in coronary arteries [57,59,60].

Alcohol and its reactive metabolites exert toxic effects virtually in every tissue inducing also a wide spectrum of disease manifestations, such as carcinogenesis [61,62,63], disturbed brain function [64,65], heart failures [66,67,68] as well as increased all-cause mortality [69,70]. Such outcomes frequently coincide with abnormalities in blood lipid profiles and the status of inflammation [71,72,73]. Current data show that in addition to liver enzymes, the lipid profiles and inflammatory markers in response to alcohol consumption are significantly modulated by physical activity, which to some extent may also occur in a gender-dependent manner. CRP, a common clinical biomarker of inflammation, seems to follow the burden of alcohol drinking and the status of physical activity in a sensitive manner. Previously, CRP changes have been established as predictors of cardiovascular morbidity even in cases where the individuals are devoid of any apparent atherosclerotic manifestations [74,75]. CRP also seems to regulate the process of inflammation [75]. It should, however, be noted that while the present data show a lowering CRP trend from sedentary to moderate or vigorous levels of physical activity, previous studies have indicated that extreme physical exertion may also stimulate inflammatory cascades and trigger adverse consequences on immune function [12,13,15,16,17,76,77,78,79,80]. Recent studies have indicated that regular physical activity can strengthen the immune system with a flattening of the dose response curve at around 500 MET minutes per week corresponding to moderate levels of activity [81]. Physical activity also decreases the risk of cognitive decline in individuals with unfavorable lifestyle factors [82,83].

The strengths of the present work are a large number of study subjects with a comprehensive assessment of the characteristics of alcohol drinking and measurements of a wide array of biomarkers with prognostic significance. The study also included separate assessments for both women and men. It should, however, be noted that direct statistical comparisons between men and women are hampered by different cut-offs for differentiating between moderate and heavy drinking or between binge and non-binge type patterns of drinking. Some limitations of the study should also be pointed out. Due to the observational and cross-sectional nature of the work and lack of follow-up data, it is difficult to assess causal inferences. The data on lifestyle factors were based on self-reports, and it is possible that recall bias or underreporting in the habits reflecting social desirability may have occurred. However, this more likely dilutes our findings than causes overestimates in the observed associations. While adherence or non-adherence to a healthy diet may also influence individual health status, detailed data on the dietary status of the participants were unfortunately not available in this work.

Taken together, our study demonstrates, however, previously unrecognized opposing relationships between physical activity and alcohol consumption, which may also prove to be of value in public health recommendations. The data also suggest a potential for using biomarker-based algorithms in the assessment of the clinical benefits of exercise-based interventions.

## Figures and Tables

**Figure 1 ijerph-19-15048-f001:**
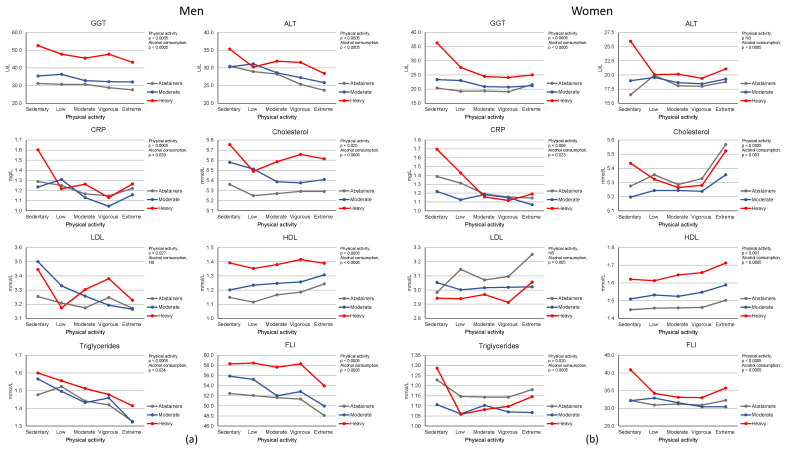
Biomarkers of liver function, inflammation, lipid status and fatty liver index in individuals with different ordered levels of physical activity and alcohol consumption. The data for liver enzymes (GGT, ALT), hs-CRP (biomarker for inflammation) and lipid profiles (cholesterol, HDL, LDL, triglycerides) are shown for both men (**a**) and women (**b**) as estimated marginal means. The subgroups for physical activity were defined as follows: sedentary activity, less than 15 min per day or no activity; low activity, 15–45 min per day; moderate activity, 45–90 min per day; vigorous activity, 90–120 min per day; extreme activity, over 120 min per day. For alcohol consumption the subgroups were as follows: abstainers = no consumption; moderate drinkers = alcohol consumption between 1 and 14 (men) or 1 and 7 (women) standard drinks per week; heavy drinkers = alcohol consumption exceeding 14 drinks (men) or 7 drinks (women) per week. The impacts of physical activity and alcohol consumption on various biomarkers were assessed using the General Linear Model including the interaction between physical activity and alcohol consumption in the model. BMI, smoking status and coffee consumption were used as covariates. *p* values for main effects of alcohol consumption and physical activity are shown. Abbreviations: GGT, gamma-glutamyltransferase; ALT, alanine aminotransferase; CRP, C-reactive protein; FLI, fatty liver index (a proxy for fatty liver), HDL, high-density lipoproteins; LDL, low-density lipoprotein.

**Table 1 ijerph-19-15048-t001:** Main characteristics of the participants, as classified according to various levels of physical activity.

	Physical Activity Level					
Men, N = 9940	Sedentary	Low	Moderate	Vigorous	Extreme	*p*-Value *
*n* (%)	681 (6.9)	917 (9.2)	5495 (55.3)	1814 (18.2)	1033 (10.4)	
Age, years, mean ± SD	47.5 ± 12.2	47.9 ± 12.7	47.9 ± 13.1	52.2 ± 13.5	53.1 ± 14.5	<0.0005
BMI, kg/m^2^, mean ± SD	28.0 ± 4.9	27.5 ± 4.6	27.0 ± 3.9	27.0 ± 3.8	26.7 ± 3.6	<0.0005
Waist circumference, cm, mean ± SD	98.6 ± 13.6	97.3 ± 12.6	95.3 ± 11.3	95.6 ± 11.1	93.7 ± 10.8	<0.0005
**Women, N = 11,110**						
*n* (%)	303 (2.7)	676 (6.1)	5971 (53.7)	3033 (27.3)	1127 (10.1)	
Age, years, mean ± SD	47.4 ± 12.0	46.8 ± 12.5	46.1 ± 12.9	48.6 ± 13.5	51.1 ± 13.4	<0.0005 **
BMI, kg/m^2^, mean ± SD	27.7 ± 6.1	27.7 ± 6.2	26.3 ± 5.1	26.1 ± 4.6	26.2 ± 4.4	<0.0005
Waist circumference, cm, mean ± SD	87.4 ± 15.4	87.0 ± 14.8	83.6 ± 13.0	83.4 ± 12.1	83.0 ± 11.7	<0.0005

** p*-value for linear trend, if not otherwise indicated. ** *p*-value for quadratic trend.

**Table 2 ijerph-19-15048-t002:** Characteristics of alcohol consumption and other lifestyle factors in individuals with various levels of physical activity.

	Physical Activity Level					
Men, N = 9940	Sedentary	Low	Moderate	Vigorous	Extreme	*p*-Value *
Alcohol intake, g/day, mean ± SD	18.9 ± 28.4	15.3 ± 22.5	13.2 ± 16.9	11.3 ± 15.3	10.6 ± 15.5	<0.0005
Average level of drinking						
abstainers, total 2721	194 (28.5%)	277 (30.2%)	1410 (25.7%)	500 (27.6%)	340 (32.9%)	0.075
moderate, total 5529	298 (43.8%)	449 (49.0%)	3135 (57.1%)	1079 (59.5%)	568 (55.0%)	<0.0005
heavy, total 1690	189 (27.8%)	191 (20.8%)	950 (17.3%)	235 (13.0%)	125 (12.1%)	<0.0005
Total 9940	681 (100%)	917 (100%)	5495 (100%)	1814 (100%)	1033 (100%)	
Drinking pattern						
regular, total 4777	268 (55.0%)	380 (59.4%)	2685 (65.7%)	948 (72.1%)	496 (71.6%)	<0.0005
binge, total 2442	219 (45.0%)	260 (40.6%)	1400 (34.3%)	366 (27.9%)	197 (28.4%)	<0.0005
Total 7219	487 (100%)	640 (100%)	4085 (100%)	1314 (100%)	693 (100%)	
Type of alcohol preferred						
wine, total 951	47 (9.7%)	67 (10.5%)	564 (13.8%)	190 (14.5%)	83 (12.0%)	0.057
beer, total 3535	249 (51.1%)	334 (52.2%)	2014 (49.3%)	616 (46.9%)	322 (46.5%)	0.010
cider/long drink, total 139	18 (3.7%)	15 (2.3%)	77 (1.9%)	19 (1.4%)	10 (1.4%)	0.003
spirit, total 1311	109 (22. %)	123 (19.2%)	692 (16.9%)	240 (18.3%)	147 (21.2%)	0.892
mixed, total 1283	64 (13.1%)	101 (15.8%)	738 (18.1%)	249 (18.9%)	131 (18.9%)	0.004
Total 7219	487 (100%)	640 (100%)	4085 (100%)	1314 (100%)	693 (100%)	
Smoking						
no, total 6930	369 (54.7%)	538 (59.1%)	3869 (70.9%)	1361 (75.2%)	793 (77.5%)	<0.0005
yes, total 2943	306 (45.3%)	373 (40.9%)	1586 (29.1%)	448 (24.8%)	230 (22.5%)	<0.0005
Total 9873	675 (100%)	911 (100%)	5455 (100%)	1809 (100%)	1023 (100%)	
cigarettes/day, mean ± SD	9.5 ± 13.0	7.3 ± 10.8	4.4 ± 8.4	3.8 ± 8.0	3.3 ± 7.3	<0.0005
Coffee, cups/day, mean ± SD	5.5 ± 4.2	4.9 ± 3.5	4.5 ± 3.0	4.5 ± 3.0	4.5 ± 3.5	<0.0005
**Women, N = 11,110**						
Alcohol intake, g/day, mean ± SD	6.4 ± 11.5	6.0 ± 9.7	5.3 ± 8.4	4.2 ± 6.6	4.1 ± 6.6	<0.0005
Average level of drinking						
abstainers, total 4707	147 (48.5%)	303 (44.8%)	2360 (39.5%)	1369 (45.1%)	528 (46.9%)	0.003
moderate, total 5099	107 (35.3%)	266 (39.3%)	2844 (47.6%)	1386 (45.7%)	496 (44.0%)	0.179
heavy, total 1304	49 (16.2%)	107 (15.8%)	767 (12.8%)	278 (9.2%)	103 (9.1%)	<0.0005
Total 11,110	303 (100%)	676 (100%)	5971 (100%)	3033 (100%)	1127 (100%)	
Drinking pattern						
regular, total 5663	126 (80.8%)	313 (83.9%)	3161 (87.5%)	1522 (91.5%)	541 (90.3%)	<0.0005
binge, total 740	30 (19.2%)	60 (16.1%)	450 (12.5%)	142 (8.5%)	58 (9.7%)	<0.0005
Total 6403	156 (100%)	373 (100%)	3611 (100%)	1664 (100%)	599 (100%)	
Type of alcohol preferred						
wine, total 2217	37 (23.7%)	110 (29.5%)	1275 (35.3%)	587 (35.3%)	208 (34.7%)	0.037
beer, total 2127	72 (46.2%)	128 (34.3%)	1198 (33.2%)	530 (31.9%)	199 (33.2%)	0.030
cider/long drink, total 413	10 (6.4%)	30 (8.0%)	224 (6.2%)	108 (6.5%)	41 (6.8%)	0.979
spirit, total 565	21 (13.5%)	45 (12.1%)	287 (7.9%)	151 (9.1%)	61 (10.2%)	0.744
mixed, total 1081	16 (10.3%)	60 (16.1%)	627 (17.4%)	288 (17.3%)	90 (15.0%)	0.731
Total 6403	156 (100%)	373 (100%)	3611 (100%)	1664 (100%)	599 (100%)	
Smoking						
no, total 8824	182 (60.3%)	480 (71.2%)	4769 (80.2%)	2462 (81.6%)	931 (82.8%)	<0.0005
yes, total 2243	120 (39.7%)	194 (28.8%)	1179 (19.8%)	557 (18.4%)	193 (17.2%)	<0.0005
Total 11,067	302 (100%)	674 (100%)	5948 (100%)	3019 (100%)	1124 (100%)	
cigarettes/day, mean ± SD	6.4 ± 9.4	4.1 ± 7.7	2.2 ± 5.3	2.0 ± 5.0	1.9 ± 5.0	<0.0005
Coffee, cups/day, mean ± SD	4.1 ± 3.2	3.8 ± 2.5	3.6 ± 2.5	3.8 ± 2.4	3.9 ± 2.6	<0.0005 **

* *p*-value for linear trend, if not otherwise indicated. ** *p*-value for quadratic trend.

## Data Availability

THL Biobank administrates and grants access to the FINRISK data to research projects that are of high scientific quality and impact, are ethically conducted, and that correspond with the research areas of THL Biobank. All data are available for application at https://thl.fi/en/web/thl-biobank/for-researchers/sample-collections/the-national-finrisk-study-1992-2012. The name of dataset is the National FINRISK Study 1992–2012. Interested researchers can replicate our study findings in their entirety by directly obtaining the data and following the protocol in the Methods section. The authors did not have any special access privileges that others would not have. More information: finriski(at)thl.fi.

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
