# Peer review of "Impact of Physical Activity on the Characteristics and Metabolic Consequences of Alcohol Consumption: A Cross-Sectional Population-Based Study"

_ijerph, 2022, doi:10.3390/ijerph192215048_

Round 1

Reviewer 1 Report

This paper investigated an interesting question based a large sample. I would like to give the following suggestions to authors to extend the contributions.

1. The title of this manuscript is too long. Readers could not catch the key points of this manuscript. Please consider to short it down.

2. Some small mistakes. For example, line 20, the authors missed a comma before we.

3. The literature part is too short and lacks up-to-date literature. For example, the authors submitted the manuscript to IJERPH. The IJERPH per se has many articles related to physical exercise (see following). Please consider to include them (as well as other latest research related to your study) in your introduction part and discussion part.  

References:

Deguchi, N.; Kojima, N.; Osuka, Y.; Sasai, H. Factors Associated with Passive Sedentary Behavior among Community-Dwelling Older Women with and without Knee Osteoarthritis: The Otassha Study. Int. J. Environ. Res. Public Health 202219, 13765. https://doi.org/10.3390/ijerph192113765

Liu, D.; Han, S.; Zhou, C. The Influence of Physical Exercise Frequency and Intensity on Individual Entrepreneurial Behavior: Evidence from China. Int. J. Environ. Res. Public Health 202219, 12383. https://doi.org/10.3390/ijerph191912383

4. I didn't observed the Figure 1A or 1B in your submitted manuscript. Please include it.

5. I didn't see GLS table in your manuscript. Please report it as you mentioned you used GLS.

6. Why GLS, not OLS?Please indicate reasons.

Author Response

This paper investigated an interesting question based a large sample. I would like to give the following suggestions to authors to extend the contributions.

  1. The title of this manuscript is too long. Readers could not catch the key points of this manuscript. Please consider to short it down.

We thank the reviewer for the positive overall assessment and useful suggestions on our work. Based on the reviewer’s recommendation, we have now sharpened the title of the manuscript as follows: “Impact of Physical Activity on the Characteristics and Metabolic Consequences of Alcohol Consumption: a Cross-Sectional Population-Based Study”.

  1. Some small mistakes. For example, line 20, the authors missed a comma before “we”.

The typographical error indicated by the reviewer has now been corrected.

  1. The literature part is too short and lacks up-to-date literature. For example, the authors submitted the manuscript to IJERPH. The IJERPH per se has many articles related to physical exercise (see following). Please consider to include them (as well as other latest research related to your study) in your introduction part and discussion part.  

References:

Deguchi, N.; Kojima, N.; Osuka, Y.; Sasai, H. Factors Associated with Passive Sedentary Behavior among Community-Dwelling Older Women with and without Knee Osteoarthritis: The Otassha Study. Int. J. Environ. Res. Public Health 202219, 13765. https:/doi.org/10.3390/ijerph192113765

Liu, D.; Han, S.; Zhou, C. The Influence of Physical Exercise Frequency and Intensity on Individual Entrepreneurial Behavior: Evidence from China. Int. J. Environ. Res. Public Health 202219, 12383. https://doi.org/10.3390/ijerph191912383

We thank the reviewer for this useful suggestions. We have now updated the manuscript by including additional citations to most recent articles in this highly active field of research (Introduction and Discussion Sections).  We also appreciate the reference to the works by Deguchi and coworkers (2022) and Liu et al (2022) which have now been included in the revised manuscript to broaden the scope of the discussion on the beneficial effects of physical activity on a wide variety of aspects related to health and wellbeing.

  1. I didn't observed the Figure 1A or 1B in your submitted manuscript. Please include it.

In the original manuscript submission Figure 1a-b was included as separate file, which appears not to have been readily accessible for the review process. Therefore, we have now attached this figure also in the main body of the revised manuscript.

  1. I didn't see GLS table in your manuscript. Please report it as you mentioned you used GLS.

We thank the reviewer for pointing this out. In the revised manuscript, we have now clarified the presentation on the use of the general linear model (GLM) in our work (Statistical methods section). GLM with ordinary least squares (OLS) was applied to assess the main and interaction effects of physical activity and alcohol consumption on various biomarkers using BMI, smoking status and coffee consumption as covariates. The results are presented in Figure 1a-b of the revised manuscript instead of a table because we feel that a graphical presentation for this type of data is more informative to the reader.

  1. Why GLS, not OLS?Please indicate reasons.

See also above (comment no 5). In our statistics, we used ordinary least squares (OLS) method, which has now been mentioned in the Methods Section.

Reviewer 2 Report

This is a very good piece of work, well presented. The topic is relevant and has both scientific and social significance. 

A few alterations below will make it more understandable to the wider audience.

Table 2 - Can you please clearly indicate the units for each parameter on the left hand side 1st column and also state what the values in brackets indicate.

I am not able to see any of the figures on the PDF. I assume that units have been appropriately stated on each graph.

Also I assume that the graphs show levels and not the risk. This is because in the discussion line 219 you mention that .. showed reduced risk. I believe you are not calculating the "Risk". Are you? This comment is based on your analyses I presume.

Any striking gender-based differences (whether or not) need to be stated, and discussed effectively and sufficiently in the Discussion 

Author Response

This is a very good piece of work, well presented. The topic is relevant and has both scientific and social significance. 

A few alterations below will make it more understandable to the wider audience.

We thank the reviewer for the positive overall assessment and useful suggestions, which have now been carefully followed in preparing the revision.

Table 2 - Can you please clearly indicate the units for each parameter on the left hand side 1st column and also state what the values in brackets indicate.

Based on the reviewer`s recommendation we have now clarified the presentation of the data in Table 2. The total numbers of observations and the units used have now been systematically given on the left hand side of the table. The numbers in brackets indicate the proportion percentages (%) of the observations in each category which has now been more clearly presented. The last column on the right specifies the p values for either linear or quadratic trends in the distribution of the observations, as indicated. The corresponding clarifications have also been made in Table 1 of the revised manuscript.

I am not able to see any of the figures on the PDF. I assume that units have been appropriately stated on each graph.

In the original manuscript submission Figure 1a-b was included as separate file, which appears not to have been readily accessible for the review process. Therefore, we have now attached this figure also in the main body of the revised manuscript.

Also I assume that the graphs show levels and not the risk. This is because in the discussion line 219 you mention that .. showed reduced risk. I believe you are not calculating the "Risk". Are you? This comment is based on your analyses I presume.

We thank the reviewer for pointing this out. In the revised manuscript we have now improved the clarity of the sentence indicated by the reviewer by using more accurate wording in referring to the findings (Discussion Section, first paragraph).

Any striking gender-based differences (whether or not) need to be stated, and discussed effectively and sufficiently in the Discussion

The reviewer raises an important point. In the revised manuscript we have now provided a more detailed account on the gender-specific characteristics of the findings (Results and Discussion Sections). Additional comparisons of biomarker responses between men and women with different levels of physical activity and alcohol consumption showed differences in LDL- and total cholesterol responses in those with the highest levels of physical activity, whereas the magnitude and direction of changes in other markers followed essentially similar patterns (Figure 1a-b). It should be emphasized, however, that any further statistical comparisons between gender-based responses are hampered by different cut-offs used for differentiating between moderate and heavy drinking or between binge and non- binge type patterns of alcohol drinking in men and women. The above views have also been covered in the Discussion Section of the revised manuscript.

Round 2

Reviewer 1 Report

Thanks for your consideration. This version is better.